# Botulinum Toxin Type A and Physiotherapy in Spasticity of the Lower Limbs Due to Amyotrophic Lateral Sclerosis

**DOI:** 10.3390/toxins11070381

**Published:** 2019-07-01

**Authors:** Riccardo Marvulli, Marisa Megna, Aurora Citraro, Ester Vacca, Marina Napolitano, Giulia Gallo, Pietro Fiore, Giancarlo Ianieri

**Affiliations:** Department of Basic Sciences, Neuroscience and Sense Organs, University of Bari “Aldo Moro”, G. Cesare Place 11, 70125 Bari, Italy

**Keywords:** amyotrophic lateral sclerosis, lower limb spasticity, botulinum toxin type A, activities of daily living, physiotherapy

## Abstract

Amyotrophic lateral sclerosis (ALS) is a progressive neurodegenerative disease (unknown pathogenesis) of the central nervous system that causes death within 1–5 years. Clinically, flabby paralysis, areflexia, muscular atrophy, and muscle fasciculations, signs of II motor neuron damage, appear. Sometimes, clinical manifestations of damage of the I motor neuron come out in lower limbs; spastic paralysis, iperflexia, and clonus emerge, and they impair deambulation and management of activities of daily living, such as personal hygiene or dressing. Thus, the first therapeutic approach in these patients involves antispasmodic drugs orally followed by botulinum toxin type A injection (BTX-A). In this study, we study the efficacy of BTX-A and physiotherapy in lower limb spasticity due to ALS and no response to treatment with oral antispastic drugs. We evaluated 15 patients (10 male and five female), with a mean age of 48.06 ± 5.2 with spasticity of adductor magnus (AM), at baseline (T0, before BTX-A treatment) and in the following three follow-up visits (T1 30 days, T2 60 days, and T3 90 days after infiltration). We evaluated myometric measure of muscle tone, the Modified Ashworth Scale of AM, Barthel Index, Adductor Tone Rating Scale, and Hygiene Score. The study was conducted between November 2018 and April 2019. We treated AM with incobotulinum toxin type A (Xeomin^®^, Merz). Spasticity (myometric measurement, Adductor Tone Rating Scale, and Modified Ashworth Scale) and clinical (Barthel Index and Hygiene Score) improvements were obtained for 90 days after injection (*p* < 0.05). Our study shows the possibility of using BTX-A in the treatment of spasticity in patients with ALS and no response to oral antispastic drugs, with no side effects. The limitation of the study is the small number of patients and the limited time of observation; therefore, it is important to increase both the number of patients and the observation time in future studies.

## 1. Introduction

Amyotrophic lateral sclerosis (ALS) is a progressive neurodegenerative disease of the central nervous system with unknown pathogenesis [1]. Its incidence is 1.5–2.5/100,000 persons per year worldwide; the only established risk factors are age, family history [2], and possibly military deployment [3]. Several known genetic changes, such as the pathological hexanucleotide repeat expansion in C9ORF72 [4,5], are causally associated with familial and sporadic ALS. Peak incidence is between 50 and 75 years old, and the ratio men to women is 3:2 [2].

ALS usually causes death within 2–4 years of diagnosis [6,7]; however, some patients could survive for a decade or more [8].

Pathogenetic hypotheses are threefold: genetic factors, oxidative stress, and glutamatergic toxicity. They cause target protein damage, such as neurofilaments and organelles (e.g., mitochondria) [9]. In addition, alteration of ribonucleic acid (RNA) processing and abnormal protein aggregation could influence a major role in the pathogenesis of ALS [10,11].

Clinically, patients show gradual involvement of motor neurons in both upper (I motor neuron), localized at the level of the cerebral cortex (UMN), and lower (II motor neuron) regions, situated in the brainstem and spinal cord (LMN). Diagnostic criteria for ALS rely on clinical (I and II motor neuron deficits in multiple body regions with a progression of symptoms) and para-clinical observations of LMN by electromyography [12].

Clinically, patients show signs of II motor neuron damage with flabby paralysis (claw hand), areflexia, muscular atrophy, and muscle fasciculations; thus, muscular weakness and atrophy appear. Dysphagia, dysarthria, and dysphonia appear when cranial nerves are involved [13]. Furthermore, I and II motor neuron loss causes fatal paralysis and death [4,14]. At lower limbs, clinical manifestations of I motor neuron damage could appear as patients show spastic paralysis, iperflexia, and clonus. These clinical symptoms can jeopardize deambulation capacity and the ability and management of personal hygiene, dressing, or other activities of daily living (ADL) [15,16].

Spasticity sometimes appears as a symptom of ALS. Muscles are contracted and they are opposed to stretching by fixing the joints in forced positions, resulting in joint pain. The first therapeutic approach in these cases is oral therapy with antispasmodic drugs (baclofen). In some cases, spasticity can also be resistant to these drugs and a source of very intense pain, fixing the limbs in positions that hinder, for example, hygiene and care of the person. Thus, botulinum toxin type A (BTX-A) injection could be an appropriate alternative to reduce spasticity. Treating spasticity can be useful to improve patients’ quality of life.

Botulinum neurotoxins (BoNTs) are produced by neurotoxigenic strains of anaerobic spores of the genus *Clostridium* (*Clostridium botulinum*, *Clostridium butyrricum*, *Clostridium barati*, and *Clostridium argentinensis*) [17,18]. BoNTs bind with high affinity to peripheral cholinergic nerve terminals. They enter into cellular cytosol (with high chain (H)) and cleave SNARE protein complexes (with low chain (L)), causing a transient flaccid paralysis due to blockage of neurotransmitter release [19,20].

BoNTs are traditionally classified into seven serotypes distinguishable with animal antisera and designated with alphabetical letters from A to G [18]. However, more recent molecular genetic analysis, including the use of next-generation sequencing techniques, led to the discovery of genes encoding for many novel BoNTs; thus, they can be grouped within an existing serotype but are characterized by different amino-acid sequences (Gene Bank and Uniprot databases). Therefore, all known antigenic properties of these variants are dubbed as subtypes and indicated with the letter of the serotype followed by a number [18,19,21], for example, for serotype A, BoNT/A1, BoNT/A2, …, BoNT/An; for serotype B, BoNT/B1, BoNT/B2, …, BoNT/Bn; etc. In addition, some chimeric BoNTs were identified and labeled accordingly: BoNT/DC, BoNT/CD, and BoNT/FA [17].

The BoNT mechanism is conveniently divided into five steps: (1) binding to nerve terminals; (2) internalization within an endocytic compartment; (3) low pH-driven translocation of the L chain across the vesicle membrane; (4) reduction of the interchain disulfide bond and release of the L chain in the cytosol; (5) cleavage of SNAREs with transient flaccid paralysis due to blockage of neurotransmitter release [17,22].

BoNT/A1 is very potent and neurospecific. It has a limited diffusion when locally injected and its action is reversible with time; thus, it is the safest and most efficacious therapeutic for the treatment of a variety of human syndromes characterized by hyperfunction of selected nerve terminals [17].

After Scott’s studies in ophthalmology, botulinum toxin clinical use continuously expanded, firstly to reduce the hypercontraction of small muscles of the eyelid and face (blepharospasm and hemifacial spasm) and larger dystonic muscles of the head and neck (cervical and oromandibular dystonia), and later to treat limb movement disorders, including use-dependent cramps (occupational dystonia) and limb spasticity [23,24,25].

In this study, we study the efficacy of BTX-A and physiotherapy in lower limb spasticity due to ALS and no response to treatment with oral antispastic drugs.

## 2. Results

Myometric mean values of muscle tone (Figure 1) at t0 (AM dx = 23.45 ± 1.33, t0 AM sn = 22.99 ± 1.11) statistically decreased at t1 (AM dx = 16.23 ± 1.16, AM sn = 17.56 ± 1.12) and t2 (AM dx = 17.45 ± 2.09, AM sn = 17.99 ± 2.01), with *p* < 0.05. At t3, no statistically significant difference was found (AM dx = 22.67 ± 1.22, AM sn = 21.19 ± 1.98), with *p* < 0.05.

Values of MAS (t0 AM dx = 3 ± 1.1, AM sn = 3 ± 1.1) also followed this trend (Figure 2), with statistical significance at t1 (AM dx = 1.3 ± 0.8, AM sn = 1.4 ± 1.0) and t2 (AM dx = 1.3 ± 0.8, AM sn = 1.4 ± 0.8), and no significance at t3 (AM dx = 2.7 ± 1.1, AM sn = 2.9 ± 0.6), with *p* < 0.05.

The baseline value of Barthel Index (t0 = 32 ± 7.7) statistically increased (Figure 3) at t1 (52 ± 5.2) and t2 (49 ± 2.8), with *p* < 0.05. At t3, no statistically significant difference was found (33 ± 2.4), with *p* < 0.05.

The baseline value of Adductor Tone Rating Scale (t0 = 3 ± 1.1) decreased in a statistically significant manner at t1 (1.3 ± 0.8) and t2 (1.3 ± 0.8), with *p* < 0.05. At t3, no statistically significant difference was found (3 ± 1.1), with *p* < 0.05 (Figure 4).

The baseline value of Hygiene Score (t0 = 3.8 ± 1.6) decreased in a statistically significant manner at t1 (1.2 ± 2.4) and t2 (1.4 ± 2.2), with *p* < 0.05. At t3, no statistically significant difference was found (3.7 ± 2.5), with *p* < 0.05 (Figure 5).

## 3. Discussion

There are few approved drugs for the treatment of ALS. The first drug approved for ALS treatment was riluzole in many countries (United States, Australia, and European countries). The Food and Drug Administration (FDA) approved edavarone (60 mg intravenous infusion per day for 10 days in alternating cycles of 14 days of suspension) as an additional treatment option for patients with ALS in May 2017. Edaravone is an antioxidant blocking the production of oxygen free radicals, believed to be responsible for nervous system damage. The approval of the drug is based on a randomized, double-blind, controlled placebo clinical trial conducted in Japan on 137 patients, lasting six months; at the end of the clinical trial, patients on edaravone therapy showed less decline in daily activities evaluated by the “ALSFRS-R scale” (*p* = 0.0013). The aim of these therapies is prolonging median survival by about 2–3 months at one year [26]. In addition, non-invasive ventilation is also thought to prolong survival [27]. Faced with the absence of any cure or any medical intervention stopping ALS progression, therapy focuses on symptomatic, rehabilitative, and palliative treatment for optimizing quality of life [28].

Nine reviews in the Cochrane library showed the effectiveness of a wide range of symptomatic treatment therapies in patients with ALS [29,30,31].

There are few clinical trials on spasticity treatment in ALS [30,32]. In clinical practice guidelines, treatment with BTX-A is not mentioned. Treatment with physical therapy is recommended, along with oral medications (baclofen and tizanidine) and intrathecal baclofen if spasticity is severe [33]. However, BTX-A treatment may be more appropriate than systemic drugs for specific goals, such as improving ADL and/or gait performance, treating specific and selected muscles. Indeed, patients of this study did not tolerate high doses of oral medications.

Spasticity is often a typical symptom of ALS. It appears when the loss of cortical motor neurons prevails over spinal motor neurons. The first therapeutic approach is oral therapy with antispasmodic drugs; spasticity can often be resistant to these drugs with very intense pain, fixing the limbs in positions that hinder, for example, hygiene and care of the person. It may be necessary to use BTX-A, injected into the muscles, to achieve paralysis-reducing spasticity. This study shows that BTX-A, normally used treating sialorrhea [34], associated with moderate physiotherapy [35], can be helpful in improving the spasticity of patients with ALS for three months at least. The treatment is safe, because we had no side effects such as general weakness and/or dysphagia. Our study shows that combined therapy of BTX-A/ physiotherapy in patients with ALS can improve the rheological parameters of the spastic muscle tissue; in this way, it makes exercise easier, slowing of the decay of motor skills, acting on both cardio-respiratory efficiency and complications from reduced joint mobility. Clinical benefits obtained from this activity relate to the physical efficiency, mood, and quality of sleep (all patients reported an increase in the number of hours of sleep). Stretching and exercises targeting joint range prevent muscle contractures and retractions of connective tissues favored by immobility; they also help contain the resulting pain. Joint mobilization should be practiced by the patient or caregiver daily. The study also proves that the specific action on the AM muscles supports their passive motion. Patients have better personal hygiene and a better personal autonomy, allowing them to achieve some maneuvers more easily, such as catheterization or hygiene of the perineal areas, thereby reducing the risk of urinary tract infection, a leading cause of death in these patients [36]. The combination therapy of BTX-A/physiotherapy also shows efficacy in the onset of fatigue. Fatigue has a multifactorial etiology [35,37], being able to achieve the progressive deficit of strength and muscle tone (which favors a mechanical strain on joints and bone segments) and/or the onset of contractures/muscle spasms from spastic hypertonia, which determine the appearance of degenerative lesions of the joints and connective tissue retraction from prolonged immobility and alterations in the microcirculation. Therefore, the reduction of muscle tone and the easier mobilization of skeletal segments of a limb, via mechanisms still poorly understood, may help in reducing the onset of fatigue [36].

BTX-A and rehabilitation allowed a reduction in muscular tone for up to three months, when a second infiltration was needed [36]. In this way, there was also an improvement in the values of the Barthel Index Scale, Adductor Tone Rating Scale, and Hygiene Score. Good values of muscular properties have an important repercussion for rehabilitative treatment [38], enabling a better mobilization of the lower limbs and preventing the establishment of muscular contractures and articular deformities requiring treatment with surgical techniques [37,38,39]; all these factors lead to an improvement in the quality of the life of the patient.

In this study, we used myometric evaluation (Myoton PRO^®^) to objectively assess muscle tone, in an objective, simple, repeatable, and non-invasive way; this device also measures muscle elasticity and stiffness. Thus, the Myoton apparatus does not properly measure the severity of spasticity; however, it tracks muscular properties (tone, elasticity, and stiffness) which are different from spasticity that can be influenced by other variables.

With appropriate dosage [40], side effects or changes of progression did not occur; this suggests that BTX-A does not interfere with the ALS course. Although the efficacy and safety of BTX-A treatment cannot be concluded for other purposes or in other patient types, we think that many patients with AM spasticity due to ALS could benefit from combined treatment of BTX-A/physiotherapy at some point of the disease.

### Limitations of the Study

The small number of patients and the limited time of observation were two limitations of this study; therefore, it is important to increase both the number of patients and the observation time in future studies.

## 4. Conclusions

Our study demonstrated the efficacy of incobotulinumtoxin A in the treatment of the bilateral adductor magnus spasticity in patients with ALS and no response to oral antispastic drugs. In the near future the clinical monitoring of patients is planned with myometric measurements after the subsequent infiltrations. We also plan to increase number of patients to obtain more information about the efficacy of BTX-A in the spasticity of these patients.

## 5. Materials and Methods

### 5.1. Study Population and Inclusion Criteria

In total, 15 patients (10 male and five female) with a mean age of 48.06 ± 5.2 with a spasticity of adductor magnus (AM) were enrolled (Table 1).

Inclusion criteria were as follows: (1) diagnosis of probable or definite ALS based on revised El Escorial Criteria (Brooks 2000); (2) presence of adductor magnus spasticity that determined a progressive disability in the activities of daily living (ADL); (3) previous and inconsequential use of oral antispastic drug therapy (Lioresal^®^ 75 mg/day) with side effects such as cough, dyspnea, and asthenia that interrupted treatment; (4) absence of antibiotic therapy during the week preceding the treatment; (5) availability of the patient to undergo continuous and intensive physiotherapy; (6) absence of fibrotic degeneration of the spastic muscles, evaluated by ultrasound.

Before participating in the study, all patients gave their informed consent via a signed form for inclusion. We conducted the study in accordance with the Declaration of Helsinki, and the protocol number of the Ethics Committee was 1626 (approved on 5 October 2018).

The study was conducted between November 2018 and April 2019.

### 5.2. Methods

Patients were evaluated before BTX-A treatment (baseline, T0) and in the three follow-up visits (T1 30 days, T2 60 days, and T3 90 days after infiltration).

Primary outcomes were to demonstrate clinical (MAS and Adductor Tone Rating Scale) and instrumental (myometric) spasticity improvement, while secondary outcomes were to demonstrate quality of living modifications (Barthel Index and Hygiene Score).

At baseline, a neurological visit was performed by a neurologist expert in motor neuron diseases, including a standard neurological exam and functional assessment through the revised Amyotrophic Lateral Sclerosis Functional Rating Scale (ALSFRS-r) and an ultrasound study of the muscles of interest to exclude the presence of fibrotic degeneration.

At baseline and during the follow-up visits, patients underwent a physical examination (patients showed independent erect standing possible only with bilateral support and for a few minutes; deambulation was possible with double support for a short time, taking little steps in a precautionary manner; spastic paraparesis resulted in a difficulty in washing and dressing due to AM spasticity) with the use of myometric measurements to evaluate muscle tone [41,42] and the following scales: Modified Ashworth Scale, Barthel Index, Adductor Tone Rating Scale, and Hygiene Score [36].

After the treatment, patients started a rehabilitation program consisting of muscular stretching, functional rehabilitation, active and passive mobilization, step training, and self-perception exercises, while advised not to exceed the hard work threshold [43]. This program was carried out three days per week.

We treated AM with incobotulinum toxin type A (Xeomin^®^, Merz Pharma); the mean dose was 99.33 ± 9.6 for right AM and 101.33 ± 11.23 for left AM via ultrasound-guided injection (Table 2).

### 5.3. Statistical Analysis

Data are expressed as means ± standard deviation. Two–way ANOVA was used to compare statistical differences of myometric measures and scale values between baseline and follow-up.

## Figures and Tables

**Figure 1 toxins-11-00381-f001:**
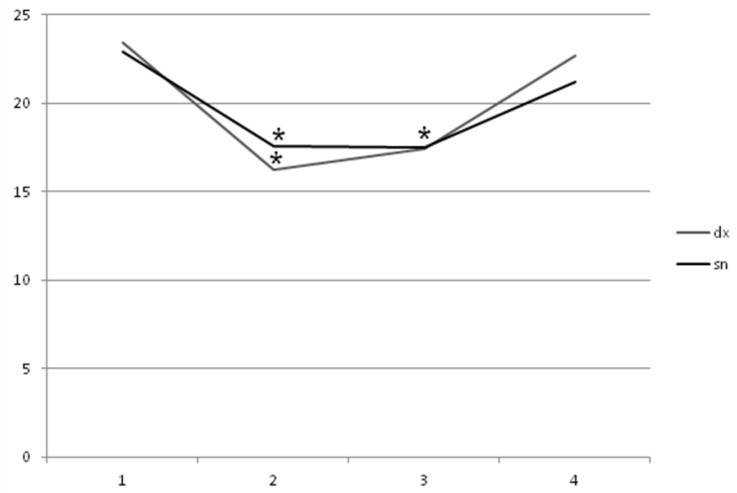
Instrumental measurements showing statistically significant decrease of right and left adductor magnus spasticity for 60 days with *p* < 0.05. AM dx: t0 = 23.45 ± 1.33, t1 = 16.23 ± 1.16, t2 = 17.45 ± 2.09, and t3 = 22.67 ± 1.22; AM sn: t0 = 22.99 ± 1.11, t1 = 17.56 ± 1.12, t2 = 17.99 ± 2.01, and t3 = 21.19 ± 1.98.

**Figure 2 toxins-11-00381-f002:**
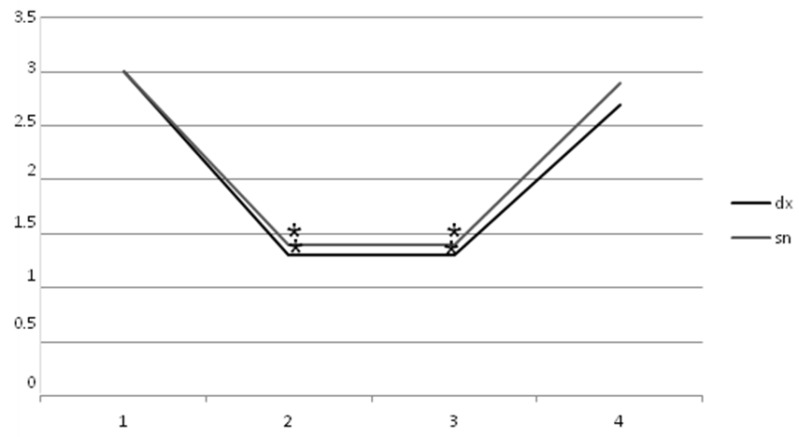
Statistically significant decrease of Modified Ashworth Scale of right and left adductor magnus during the study (*p* < 0.05). AM dx: t0 = 3 ± 1.1, t1 = 1.3 ± 0.8, t2 = 1.3 ± 0.8, and t3 = 2.7 ± 1.1. AM sn: t0 = 3 ± 1.1, t1 = 1.4 ± 1.0, t2 = 1.4 ± 0.8, and t3 = 2.9 ± 0.6.

**Figure 3 toxins-11-00381-f003:**
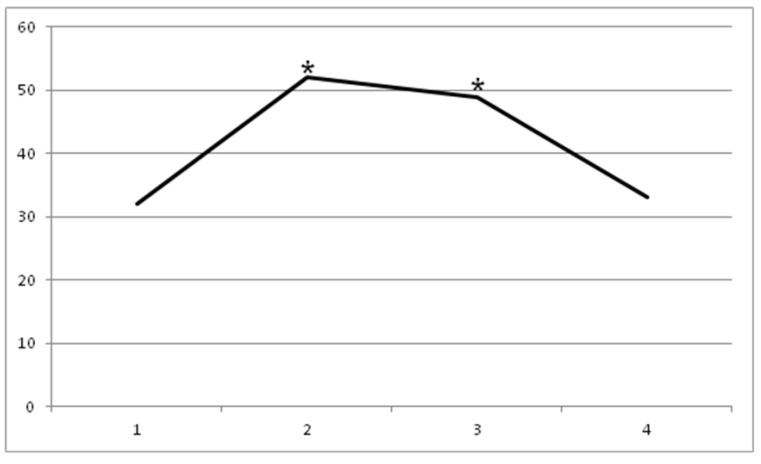
Barthel Index increase until 60 days (*p* < 0.05): t0 = 32 ± 7.7, t1 = 52 ± 5.2, t2 = 49 ± 2.8, and t3 = 33 ± 2.4.

**Figure 4 toxins-11-00381-f004:**
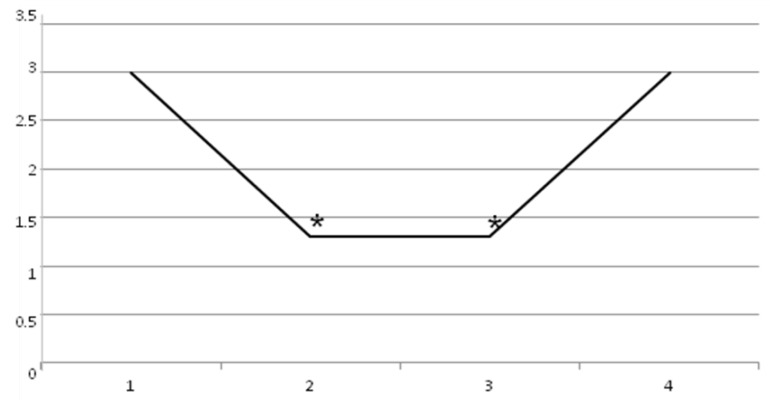
Adductor Tone Rating Scale decrease until 60 days (*p* < 0.05): t0 = 3 ± 1.1, t1 = 1.3 ± 0.8, t2 = 1.3 ± 0.8, and t3 = 3 ± 1.1.

**Figure 5 toxins-11-00381-f005:**
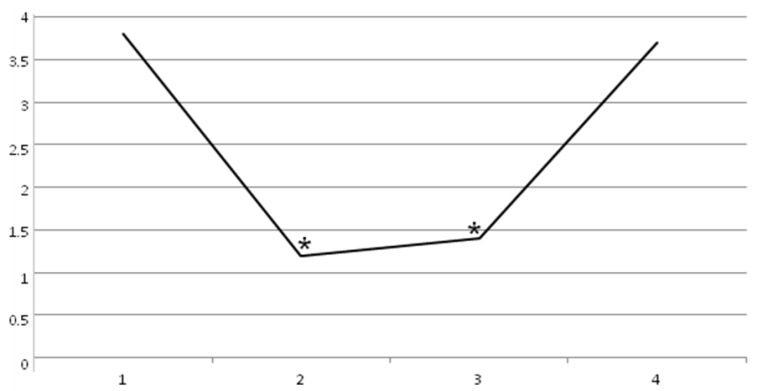
Statistically significant decrease of Hygiene Score during the study (*p* < 0.05): t0 = 3.8 ± 1.6, t1 = 1.2 ± 2.4, t2 = 1.4 ± 2.2, and t3 = 3.7 ± 2.5.

**Table 1 toxins-11-00381-t001:** This table shows the age, sex, diagnosis date, and bulbar symptom data of all patients. Just two patients (7 and 13) had bulbar signs characterized by an important dysarthria. Furthermore, 7/15 patients (46.6%) had diagnosis in 2017, while 4/15 (26.6%), 3/15 (20%), and 1/15 (6%) had diagnoses in 2016, 2018, and 2015, respectively. Pt = patient; M = male; F = female; Dg = diagnosis date; BS = bulbar signs.

	Pt 1	Pt 2	Pt 3	Pt 4	Pt 5	Pt 6	Pt 7	Pt 8	Pt 9	Pt 10	Pt 11	Pt 12	Pt 13	Pt 14	Pt 15
**Age**	55	45	45	50	55	45	46	58	45	45	47	47	55	43	40
**Sex**	M	M	M	M	F	M	F	F	M	F	F	M	M	M	M
**Dg**	2017	2016	2017	2017	2018	2016	2017	2017	2017	2015	2018	2018	2017	2016	2016
**BS**	No	No	No	No	No	No	Yes	No	No	No	No	No	Yes	No	No

**Table 2 toxins-11-00381-t002:** This table shows dosage for each muscle for each patient. Patients 1, 2, 3, and 14 showed higher spasticity; thus, dosage in these patients was higher than in others. Pt = patient; R AM = right adductor magnus; L AM = left adductor magnus.

	Pt 1	Pt 2	Pt 3	Pt 4	Pt 5	Pt 6	Pt 7	Pt 8	Pt 9	Pt 10	Pt 11	Pt 12	Pt 13	Pt 14	Pt 15
R AM	110	120	100	80	100	100	80	100	100	100	100	100	100	100	100
L AM	110	120	120	80	100	100	80	100	100	100	100	100	100	110	100

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
