# Peer review of "Botulinum Toxin Type A and Physiotherapy in Spasticity of the Lower Limbs Due to Amyotrophic Lateral Sclerosis"

_toxins, 2019, doi:10.3390/toxins11070381_

Round 1

Reviewer 1 Report

In this clinical paper, the authors investigate the use of BTX-A in the treatment of spasticity in patients with Amyotrophic Lateral Sclerosis (ALS) no responders to oral antispastic drugs.

Prior publication both major and minor point should be addressed:

1) Please revise the abstract section. Some typos are present: lane 5 “an”.

2) Please provide in the introduction section an overview about botulinum neurotoxins, their mechanism of action and their clinical use (Pirazzini M., Toxicon 2018; Azarnia Tehran D. Toxins, 2019; Pirazzini M., Pharmacol Rev. 2017).

3) Provide SEM or SD in all graphs.

4) Provide for clarity a table with the raw data for all the 15 patients. This table should indicate, sex, age, degree of disease, and all the parameters analyzed in all figures.

5) Conflicts of interest should be indicated.

Author Response

Dear Reviewer,

Thank you for your suggestions and comments. Our replies:

1) we revised the abstract section;

2) we inserted in the introduction section an overview about botulinum neurotoxins, their mechanism of action and their clinical use (line 67-96);

3) we inserted SD in all graphs;

4) we inserted two tables, one with sex, age and degree of disease (line 218) and one with botulinum toxin A dosage (line 257);

5) we indicated no conflicts of interest (line 269-270)

Reviewer 2 Report

The aim of the study was to demonstrate the efficacy of BTX-A in patients with ALS and spasticity of lower limbs who were no responders to systemic antispastic drugs.

The introduction is too long and redudant and should remodulated. The authors discuss too much about the pathogenesis of ALS and this is out of the aim of the paper.

Methods: it is not reported which are the primary outcome and secondary outcomes of the study (myometric measures ? MAS ? Quality of living ?.

The myoton apparatus do not measure the spasticity severity, but the muscular tone which is different from spasticity and may be influenced by other variables. This should be reported in the methos as well as in the discussion.

Did the author perform needle EMG in the Adductor muscles before treating the patients with BTX-A injections. The presence of fibrillation potentials -that indicates acute denervation- in these muscles may indicate the presence of contemporary LMN involvement at this level. It has been demonstrated in ALS patients with bulbar muscle involvement that the association of LMN involvement with muscle hyperactivity at the same muscular level is not effective to treat muscular hyperactivity.

Author Response

Dear Reviewer,

Thank you for your suggestions and comments. Our replies:

1)    We remodulated introduction; according to other comments of another reviewer, we inserted in this section an overview about botulinum neurotoxins, their mechanism of action and their clinical use (line 67-96);

2)    We reported in Methods section which are primary and secondary outcomes of the study (line 261-263);

3)    We specified myoton apparatus does not measure the spasticity severity, but the muscular tone (line 247 and line 192-196);

4)    We just performed needle EMG in the adductor muscles of two patients with bulbar signs (patients 7 and 13, see table 1, line 218); no fibrillation potentials (indicated acute denervation) were demonstrated.

Round 2

Reviewer 1 Report

The authors fully addressed my comments

Author Response

Dear Reviewer,

We are happy to fully address your comments.

We revised English language and style.

Best regards

Reviewer 2 Report

In the revised version of the manuscript included the primary and secondary outcomes of the study. This should be put at the beginnig of the "Methods" paragraph and not at the end.

Author Response

Dear Reviewer,

Thank you for your suggestions and comments. Our replies:

1)    We put primary and secondary outcomes of the study at the beginnig of the "Methods" paragraph and not at the end (line 238-240);

2)    We improve English language and style.

Best regards
